# Mcl-1 Protein and Viral Infections: A Narrative Review

**DOI:** 10.3390/ijms25021138

**Published:** 2024-01-17

**Authors:** Zbigniew Wyżewski, Justyna Stępkowska, Aleksandra Maria Kobylińska, Adriana Mielcarska, Matylda Barbara Mielcarska

**Affiliations:** 1Institute of Biological Sciences, Cardinal Stefan Wyszyński University in Warsaw, Dewajtis 5, 01-815 Warsaw, Poland; 2Institute of Family Sciences, Cardinal Stefan Wyszyński University in Warsaw, Dewajtis 5, 01-815 Warsaw, Poland; j.stepkowska@uksw.edu.pl; 3Division of Immunology, Department of Preclinical Sciences, Institute of Veterinary Medicine, Warsaw University of Life Sciences—SGGW, Ciszewskiego 8, 02-786 Warsaw, Poland; aleksandra.kobylinska00@gmail.com (A.M.K.); matylda_mielcarska@sggw.edu.pl (M.B.M.); 4Department of Gastroenterology, Hepatology, Nutritional Disorders and Pediatrics, The Children’s Memorial Health Institute, Av. Dzieci Polskich 20, 04-730 Warsaw, Poland; a.mielcarska@ipczd.pl

**Keywords:** apoptosis, MCL-1, HBV, HCV, HIV, EBV, KHSV, flaviviruses, IAV, SARS-CoV-2

## Abstract

MCL-1 is the prosurvival member of the Bcl-2 family. It prevents the induction of mitochondria-dependent apoptosis. The molecular mechanisms dictating the host cell viability gain importance in the context of viral infections. The premature apoptosis of infected cells could interrupt the pathogen replication cycle. On the other hand, cell death following the effective assembly of progeny particles may facilitate virus dissemination. Thus, various viruses can interfere with the apoptosis regulation network to their advantage. Research has shown that viral infections affect the intracellular amount of MCL-1 to modify the apoptotic potential of infected cells, fitting it to the “schedule” of the replication cycle. A growing body of evidence suggests that the virus-dependent deregulation of the MCL-1 level may contribute to several virus-driven diseases. In this work, we have described the role of MCL-1 in infections caused by various viruses. We have also presented a list of promising antiviral agents targeting the MCL-1 protein. The discussed results indicate targeted interventions addressing anti-apoptotic MCL1 as a new therapeutic strategy for cancers as well as other diseases. The investigation of the cellular and molecular mechanisms involved in viral infections engaging MCL1 may contribute to a better understanding of the regulation of cell death and survival balance.

## 1. Introduction

Myeloid cell leukemia-1 (MCL1) is a member of the B-cell lymphoma 2 (BCL2) family of proteins and a regulator of apoptosis. MCL1, similarly to BCL2 and BCL-xL, sequesters pro-apoptotic Bcl-2 homology (BH)3 proteins, which inhibits the activation of pro-apoptotic proteins BAX and BAK, prevents the permeabilization of the outer mitochondrial membrane (OMM), and the release of cytochrome c and consequently inhibits apoptosis [1], as presented in Figure 1.

As the antiapoptotic member of the Bcl-2 family, MCL-1 contains four BH domains, BH1–4. The BH core of this protein consists of eight α-helices. Six of these (2–5 and 8) form the hydrophobic BH3-binding pocket, a structure involved in the interaction between MCL-1 and other BCL-2 family representatives, including BAK [2]. The N-terminal part of MCL-1 contains multiple sites of chemical modifications (i.e., ubiquitination, phosphorylation, and cleavage) determining the activity, degradation, dimerization, and intracellular distribution of this protein. A part of the N-terminus is rich with the following amino acid residues: Pro (P), Glu (E), Ser (S), and Thr (T), and is therefore called the PEST region. The PEST motifs are present within proteins characterized by their short half-life [3]. The transmembrane domain (TM), a structure responsible for anchoring Mcl-1 into the lipid membranes (especially the OMM), is present at the C-terminus of the MCL-1 molecule [4].

Various studies have proven that the levels of this prosurvival protein, taken into account together with clinicopathological characteristics, have prognostic value for patients with cancer, and MCL1 may be a prognostic biomarker for patients with certain types of cancer [5]. Hence, the inhibition of MCL1 protein–protein interactions has been analyzed in many studies. A meta-analysis based on data from 2208 patients with cancer showed that high MCL1 expression was frequently observed in patients with poorly differentiated digestive system tumors or lung adenocarcinoma, and higher MCL1 expression was associated with overall shorter survival in patients with hematological and digestive system tumors or with lung cancer. Interestingly, high MCL1 expression was observed in patients with high T stage, M stage, and TNM stages in some malignancies [5]. Also, the amplification of *MCL1* can be observed in some patients with pancreatoblastoma (PB). This points to MCL1 as a possible target in PB; the inhibition of MCL1 in patients with *MCL1*-amplified cancer could be beneficial [6]. MCL1 is often upregulated in acute myeloblastic leukemia (AML) cells, which are often resistant to treatment with BCL2 inhibitors; however, the administration of an MCL1 inhibitor together with a BCL2 inhibitor leads to apoptosis in AML cells or animal models of AML [7]. Combined BCL2 and MCL1 targeting is also used against small-cell lung cancer (SCLC) when the cells show high BCL-2 expression and detectable BAX [8]. Recent AML-related studies have demonstrated that MCL1 interacts with Hexokinase 2 (HK2) on the outer mitochondrial membrane, inducing glycolysis and mitochondrial oxidative phosphorylation (OXPHOS) and thereby endowing cell metabolic plasticity, which may promote resistance to therapy [9].

Inhibiting the interaction of MCL1 with other proteins to move toward apoptosis presents significant challenges and may require small-molecule conformational restriction. Taking this knowledge for the starting point, AMG 176 was discovered as the first selective MCL1 inhibitor studied in humans in 2018 [10]. AMG 176 caused reductions in B cells, monocytes, and neutrophils in human MCL1 knock-in mice, and may provide therapeutic promise in models of AML as well as other hematologic malignancies. In another hematological neoplasm, mantle cell lymphoma (MCL), casein kinase 2 (CK2) has been shown to be over-expressed, and targeting CK2 reduces MCL-1 levels, which in turn sensitizes MCL cells to BCL-2-specific venetoclax inhibitor and presents a novel therapeutic strategy for MCL patients [11]. Interestingly, the mechanism of action of some MCL1 inhibitors used in hematological malignancies is the induction and increase in Mcl1 protein stability [12]. This occurs by blocking Mcl1 ubiquitination via the enhanced de-ubiquitination and dissociation of Mcl-1 from Noxa, Bak, and Bax.

Targeting MCL1 inhibition in breast cancer-associated fibroblasts (bCAFs), which typically express MCL1, changes their phenotype and reverses their pro-invasive properties [13]. Bonneaud et al. demonstrated that the inhibition of MCL 1 serves throughout cancer ecosystems, contributes to the death of cancer cells, and may also reverse the tumorigenic activation of bCAFs that co-evolve with cancer cells.

Interestingly, MCL-1 regulation is observed and serves as a key event during infections with various types of viruses, which is presented in detail in the following sections. In light of scientific research, Mcl-1 may determine the effectiveness of viral replication and dissemination, affect the host immune response, favor the persistence of infection, and influence the development of virus-driven diseases and pathological processes. Therefore, resultful antiviral strategies are needed, and MCL-1 should be considered a target for antiviral agents.

## 2. MCL-1 in HBV Infection

Hepatitis B virus (HBV) belongs to the *Hepadnaviridae* family and the *Orthohepadnavirus* genus [14]. Its genome has the form of partially double-stranded DNA with a size of about 3.2 kb. The genome consists of four open reading frames that overlap each other and encode the viral polymerase (P), core, surface antigens, and X protein [14,15]. During infection, DNA is converted into a covalently closed circular DNA (cccDNA) [16]. HBV is classified into at least ten genotypes named A–J [1]. Its genome is covered by a capsid composed of 240 copies of the HBV core protein. The lipid envelope encircles the capsid and contains three HBV surface antigens (HBsAg)—large (L), middle (M), and small (S) [17]. It is estimated that 250 million people live with chronic HBV infection [18]. Infection can cause a wide spectrum of liver diseases from acute hepatitis and fulminant hepatic failure to chronic hepatitis, cirrhosis, and hepatocellular carcinoma (HCC) [19].

Research has suggested that the MCL-1 protein plays an important role in maintaining the viability of malignant cells during HCC development [20,21,22]. HCC is linked with a substantial decrease in the level of a set of microRNAs including miR-29a/b/c (miR-29a, miR-29b, and miR-29c), miR-101, and miR-193b. Meanwhile, all of those molecules are responsible for the regulation of *MCL1* gene expression [20,21,22]. Su et al. [22] have performed in silico analyses to reveal the putative ability of miR-101 to bind the 3′ untranslated region (3′-UTR) of *MCL1* mRNA. The presence of complementary fragments within miR-101 and its target, respectively, suggested that the former molecule might counteract the translation of the *MCL1* gene, and thus decrease the viability of the cell by opening the mitochondrial gateway to apoptosis. Next, genetic experiments have confirmed the bioinformatic prediction. The team used plasmids with the luciferase reporter gene and the *MCL1* gene fragment encoding the miRNA-target site within the mRNA 3′-UTR, either the wild-type sequence or the mutant one. In the second case, the increase in relative luciferase activity was observed, proving the miR-101-dependent downregulation of *MCL1* gene expression. Thus, this study revealed the important mechanism that confers cancer resistance to apoptosis. miR-101 interferes with the expression of the *MCL1* gene at the posttranscriptional level. In HCC tissues, the loss of miR-101 abolishes this effect [22]. 

Further studies by Xiong et al. [20] have enabled the identification of other miRNA molecules that are engaged in the regulation of the intracellular level of MCL-1 protein in the context of HCC. miR-29a/b/c were found out to regulate the expression of the two cellular genes encoding the MCL-1 and BCL-2 proteins, respectively. miR-29a/b/c directly target the transcripts of the *MCL1* and *BCL2* genes. The use of real-time reverse transcription PCR (qRT PCR) and Northern blotting techniques allowed the researchers to reveal the substantial deficiency of miR-29a/b/c in HCC tissues. Further experiments determined the role of miR-29a/b/c in maintaining malignant cell viability under various stress conditions. The transfection of HCC cells with miR-29 duplexes sensitized the transfectants to cell death induced by the following stimuli: hypoxia, serum deprivation, or the set of proapoptotic chemotherapeutic agents including doxorubicin, curcumin, and etoposide. On the contrary, the introduction of anti-miR-29 molecules (anti-miR-29a and anti-miR-29b) conferred cell resistance to either hypoxia-induced or serum starvation-stimulated apoptosis. Interestingly, the team reproduced the proapoptotic effects of miR-29a/b/c upregulation, performing the siRNA-dependent silencing of either the *MCL1* or *BCL2* gene. Thus, the HCC cell resistance to apoptosis turned out to be the result of the HCC-dependent abolition of the miR-29a-mediated downregulation of the *MCL1* and *BCL2* mRNA translation [20]. 

Moreover, Xiong et al. [20] compared the observations of the miR-29 levels in HCC tissues with the clinical and prognostic data of HCC patients. The analysis suggested that the decrease in the amount of miR-29 is the molecular phenomenon that contributes to HCC pathogenesis. The deficiency in miR-29 was associated with an elevated level of alpha-fetoprotein (AFP) and shorter disease-free survival (DFS). This correlation suggested the important role of the miR-29 targets, the *MCL1* and *BCL2* mRNAs, and their products, the MCL-1 and BCL-2 proteins, respectively, in HCC development [20]. 

Mao et al. [21] have determined the molecular mechanism co-responsible for the development of HBV-associated HCC. The team used the Hep-G2/2.2.15 human hepatoblastoma cell line as a model system for studying HBV infection in malignant hepatocytes [21]. Due to the stable transfection with a double-dimeric vector of HBV, Hep-G2/2.2.15 cells can express the viral genes and thus reconstruct the molecular events specific to HBV-associated HCC [23]. The Western blot analysis showed the relevantly elevated level of MCL-1 protein, compared to HBV-nonexpressing cells, either HepG2 cell line or normal, nonmalignant hepatocytes. The immunohistochemical analysis confirmed the influence of HBV on the amount of MCL-1 protein. The team examined the patient-derived specimens of HCC tissues, and the HBV-infected ones displayed a higher level of MCL-1 than the uninfected ones. The real-time PCR (qPCR) data analysis revealed a significant decrease in miR-193b in Hep-G2/2.2.15 cells. Similarly, HBV lowered the miR-193b level in HCC tissues. This result suggested that HBV may upregulate the MCL-1 via disturbing the miRNA-dependent mechanism aimed against *MCL1* mRNA and thus maintains the viability of HBV-positive malignant hepatocytes. The high level of m-R193b was associated with a prolonged overall survival (OS). The manipulation of the cell regulatory machinery for *MCL1* gene expression seemed to determine the disease progression, the effectiveness of the treatment, and the prognosis for HCC patients [21].

On the other hand, in some circumstances, HBV infection may also exert the opposite effect on the MCL-1 level. Hu et al. [24] have reported that upon exposure to an anticancer agent, cisplatin, HBV-infected hepatocytes displayed a decreased level of MCL-1. The HBV HBx protein was responsible for MCL-1 downregulation. The team used two cell lines: HepG2-HBx and HepG2-con. The first one was obtained by the transfection of HepG2 cells with the plasmid encoding HBx protein, whereas the second one carried an empty vector. The cisplatin treatment caused a decrease in the intracellular level of MCL-1 in both transfectants. However, this effect was significantly stronger in the HepG2-HBx cells, suggesting the synergy between the tested drug and HBx protein in lowering the amount of MCL-1. Upon the treatment with cisplatin, the HepG2-HBx cells displayed an elevated apoptosis rate compared to the HepG2-con cells. The transfection of the five cell lines (three HCC ones and two nonmalignant hepatocyte ones) with the HBx-encoding adenoviral vector confirmed the role of the HBV HBx in enhancing the proapoptotic effect of the cell exposure to cisplatin, including the downregulation of MCL-1 protein. Further investigations revealed that HBx affects MCL-1 at the posttranslational level, and the loss of MCL-1 in HBx-producing cisplatin-stimulated cells is the effect of the increased rate of protein degradation. This effect was associated with oxidative stress and the intensive production of reactive oxygen species (ROS). Moreover, the MCL-1 downregulation turned out to occur in a caspase-3-dependent manner. The exposure to antioxidants, either N-acetylcysteine (NAC) or glutathione (GSH), counteracted the cisplatin-induced downregulation of MCL-1 in both the HepG2-HBx and Hep-G2/2.2.15 cells, rendering them resistant to apoptosis. The use of a pan-caspase inhibitor, Z-VAD-FMN or caspase-3 inhibitor, AC-DEVD-CHO, caused similar results [24].

Interestingly, in vivo studies have shown the HBx-mediated enhancement of cisplatin-dependent MCL-1 downregulation worsens liver condition. Mice transfected with a plasmid encoding HBx protein as well as control wild-type animals were treated with the DNA construct carrying the *MCL1* gene or with the empty vector. Thus, the set of transfectants included HBx-positive individuals producing an extra amount of MCL-1, the mice able to synthesize either HBx or additional MCL-1, and the ones lacking HBx and surplus MCL-1. Finally, MCL-1 was found to perform a hepatoprotective function. Among the HBx-positive transfectants treated with cisplatin, the ones overproducing MCL-1 displayed decreased levels of two enzymatic biomarkers of liver injury: alanine aminotransferase (ALT) and aspartate aminotransferase (AST), in comparison to the mice nontransfected with MCL-1-encoding vector [24]. 

Thus, MCL-1 plays a dual role in the context of HBV infection. As previously mentioned, the upregulation of this protein may contribute to HCC pathogenesis [a]. On the contrary, MCL-1, together with another antiapoptotic member of the BCL-2 family, BCL-XL, have been found to play an essential role in maintaining the homeostasis, proper functioning, and integrity of the liver [cis, HIK]. The study by Hu et al. [24] suggests that the HBx-dependent loss of MCL-1 decreases the viability of the hepatocytes, and intensified cell death leads to liver injury. Therefore, either the upregulation or downregulation of MCL-1 may contribute to the development of HBV-driven disorders and diseases [24].

## 3. MCL-1 in HCV Infection

Hepatitis C virus (HCV) belongs to the *Flaviviridae* family and the *Hepacivirus* genus [25]. Its genome has the form of positive-sense single-stranded RNA (ssRNA) [26] and has one large open reading frame flanked by 5′ and 3′ untranslated regions (3′-UTR and 5′-UTR). The translation leads to the formation of a single polyprotein containing structural proteins such as a core protein, two envelope glycoproteins, and six non-structural proteins including viral RNA-dependent RNA polymerase (vRdRp) [27]. HCV infection is a serious global health problem. Chronic infection can cause liver disease including cirrhosis and HCC. It is estimated that 170 million people are chronically infected and three to four million persons are infected each year [28].

The HCV infection implies the complex interplay between the pathogen and the host’s innate and adaptive immune response. The second arm of the host defense includes the CD8+ cytotoxic T lymphocytes (CTLs). As a result of activation and differentiation into functional effector cells, CTLs gain competencies to eliminate the virus, killing directly the infected cells. However, long-term exposure to viral antigens attenuates the CTL response. Persistent HCV infection affects the reactivity of CTLs and makes them unable to destroy the target cells. The substantial mitigation of the CTL response contributes to the development of chronic hepatitis C. CTLs may adapt to fight the HCV through the surface expression of the interleukin (IL)-7 receptor, also known as the cluster of differentiation (CD)127. However, the percentage of CD127+ cells in the population of CD8+ lymphocytes decreases during chronic HCV infection. Thus, an effective treatment is required to prevent this effect and retain the IL-7-sensitive CTLs, making the host competent at eliminating the pathogen [29,30,31,32]. 

IL-7 signaling determines the CTL viability. The surface expression of CD127 promotes cell survival after the stimulation by the viral antigen [29,31]. Larrubia et al. [31] characterized the role of MCL-1 in CD127+ CTLs. MCL-1 and another BCL-2 family member—BIM—determined the apoptotic and proliferative potential of antigen-stimulated CTLs. The team isolated CTLs from the HCV-infected patients. CD8+ T lymphocytes able to recognize the fragments of the HCV NS3 protein were divided into two categories. The first one exhibited the high cell surface expression of CD127 (CD127high CTLs), whereas the second one displayed the low level of the surface CD127low CTLs. Upon antigen stimulation, the CD127high CTLs were characterized by a higher proliferative potential than CD127low lymphocytes. This difference is reflected in the effectiveness of the host immune response and the severity of HCV-driven hepatitis. In patients with CD127high CTLs, the viral propagation was substantially less efficient, which was manifested by the low or even undetectable HCV viral load. The weak surface expression of the CD127 resulted in a low level of alanine aminotransferase (ALT). In the individuals with CD127low, both the viral load and the amount of the aforementioned enzyme marker were high. Further in vitro analysis revealed the positive correlation between the cell-surface expression of CD127 and the intracellular amount of MCL-1 in the antigen-stimulated HCV NS3-specific CTLs. The opposite relationship existed between the levels of surface CD127 and BIM. These findings suggest that chronic HCV infection may disrupt IL-7/CD127-mediated signaling and lower the MCL-1/BIM ratio, thereby decreasing the apoptotic potential, proliferative ability, and reactivity of the antigen-stimulated CTLs [31].

Previous research has identified the viral product that could target MCL-1 directly [33]. Mohd-Ismail et al. [33] have studied the structure of the HCV core protein to reveal the sites involved in the interaction with MCL-1. As evidenced by the analysis, three hydrophobic amino acid residues, Leu119, Val122, and Leu126, within the BH3-like domain of the viral molecule, were responsible for binding and blocking MCL-1. The MCL-1 targeting led to cell death. The transfection of the Huh7 cells with the core protein-encoding vector increased the apoptosis level. Meanwhile, this effect did not occur in the transfectants, producing incomplete core molecules, i.e., without a BH3-like domain [33].

## 4. MCL-1 in HIV Infection

Human immunodeficiency virus (HIV) is a member of the *Retroviridae* family and the *Lentivirus* genus [34]. Its genome is in the form of positive-sense single-stranded RNA [35], and is enclosed in an enveloped, cone-shaped capsid that is 100–150 nm in diameter [36]. Each viral particle contains two copies of the full-length viral genomic RNA [37,38]. Each copy encodes at least fifteen proteins including structural, envelope, and accessory proteins, and enzymes such as reverse transcriptase (RT) [39,40]. During infection, the viral genome undergoes reverse transcription into double-stranded DNA (dsDNA). Viral RT uses the viral RNA as a template to synthesize the first strand and then uses the resulting complementary DNA to synthesize the second strand. Integrated with the host genome, the viral dsDNA is a template for viral particles [41]. HIV is the etiological agent of acquired immunodeficiency syndrome (AIDS) [42]. Many infected people are unaware of the infection because the infection is asymptomatic or has flu-like symptoms for 5 to 10 years [43].

The HIV’s ability to hide inside the immune cells, i.e., CD4+ lymphocytes and monocytes/macrophages, determines the persistence of viral infection, making the pathogen resistant to the host immune response, thwarting antiviral therapies, and contributing to systemic spread and transmission. Infected cells act as the reservoirs of HIV; therefore, the mechanisms maintaining their viability play an essential role in the virus–host interplay [44,45,46,47]. 

HIV can infect monocytes/macrophages without causing either programmed cell death or significant cytopathic effects [45,46,47]. Busca et al. [44] have examined the influence of the MCL-1 protein on the apoptotic potential of these cells. The team used a human monocytic cell line: THP1. It was stimulated with phorbol-12-myristate-13-acetate (PMA) to induce the specialization into THP1-derived macrophages (THP1-MACs) and thus obtain the monocyte-to-macrophage differentiation model. The authors also treated human peripheral blood mononuclear cell (PBMC)-derived primary monocytes with macrophage colony-stimulating factor (M-CSF) to produce monocyte-derived macrophages (MDMs). The specialization of THP1 cells into THP1-MACs was associated with the elevation of the level of the MCL-1 protein. The second differentiation model displayed similar results: the expression of the *MCL1* gene was significantly higher in the MDMs than in the primary monocytes. Moreover, the PMA-dependent or M-CSF-dependent increase in another member of the BCL-2 family, BCL-XL, was observed in the THP1-MACs and MDMs, respectively. These results suggested that monocyte-to-macrophage differentiation enhances the viability of the HIV-infected macrophages by affecting the intracellular mitoprotective signaling, and thus preventing the intrinsic apoptosis pathway. The level of prosurvival members of the BCL-2 family was analyzed in the context of the macrophage resistance to viral protein R (Vpr) [44]. Vpr is an apoptotic regulator that can affect the host cell viability via binding to adenine nucleotide translocator (ANT), a protein spanning the inner mitochondrial membrane (IMM) [48]. Vpr–ANT interaction may lead to the permeabilization of mitochondria as well as IMM depolarization, which causes the consequent leakage of proapoptotic factors out of the mitochondrial intracellular space [49]. Busca et al. [44] have determined the influence of the monocyte-to-macrophage specialization on cell insensitivity to Vpr-induced apoptosis. In light of the obtained results, the maturation of the cells implicates the substantial increase in the resistance to the proapoptotic activity of the synthetic C-terminal fragment (52–96 aa) of Vpr. Under the treatment with this peptide, the THP1-MACs and MDMs displayed meaningly reduced levels of apoptosis compared to the THP1 and PMBC-derived primary monocytes, respectively. Research has shown that Vpr (represented by a functional C-terminal fragment) causes the substantial loss of MCL-1 in THP1 cells, whereas it does not exert a significant influence on the MCL-1 level in THP1-MACs. The silencing of the MCL-1 expression with MCL-1-specific small interfering RNA (siRNA) enabled the examination of the possible antagonism between MCL-1 and Vpr in the regulation of the cell viability. The lack of MCL-1 resulted in the spontaneous apoptosis of the THP1-MACs. However, the treatment of the MCL-1-deficient THP1-MACs with a C-terminal Vpr fragment did not elevate the level of cell death (compared to MCL-1-specific siRNA-untreated cells), indicating that the prosurvival role of MCL-1 is not based on the antagonistic relationship between MCL-1 and Vpr [44]. The findings of Busca et al. [44] revealed that the expression of the *MCL1* gene co-determines the constitutive viability of macrophages and thus indirectly contributes to the persistence of HIV infection [44].

Swingler et al. [50] have studied the molecular mechanism responsible for the elevation of the MCL-1 level in HIV-infected human macrophages. The research linked the upregulation of MCL-1 with the activity of M-CSF, whereas the production of this cytokine was stimulated by the HIV envelope glycoprotein (Env). Macrophages infected with the wild-type HIV-1 displayed an elevated level of two mitoprotective factors, MCL-1 as well as another representative of the BCL-2 protein family, BFL-1. The treatment of the mock-infected cells with M-CSF had a similar effect. On the other hand, the HIV-1 mutant lacking the *env* gene (HIV Δenv) invaded macrophages without affecting the level of either MCL-1 or BFL-1. The significant effect of the Env-dependent upregulation of MCL-1 and BFL-1 was an increase in the cell resistance to apoptosis mediated by tumor necrosis factor (TNF)-related apoptosis-inducing ligand (TRAIL) [50]. TRAIL is well known as the protein that induces an extrinsic pathway of apoptosis. It interacts with TRAIL receptors (TRAILRs), the cell surface molecules that span across the plasma membrane and are responsible for the transmission of the proapoptotic signal from the extracellular messenger to the cell interior. TRAIL signaling has been studied in the context of malignancies, such as melanoma [51], cervical cancer, lung carcinoma [52], breast cancer [53], cholangiocarcinoma [54], leukemia [55], and colorectal cancer [56]. In light of tumor research, the MCL-1 protein determines the phenomenon of cell resistance to TRAIL-induced apoptosis [51]. For example, Sarif et al. [51] have revealed that the siRNA-dependent silencing of the *MCL1* gene as well as the MCL-1 protein with the use of S63845 increased BAX and BAK activation, and sensitized human melanoma cells to the proapoptotic activity of TRAIL [51]. Therefore, the results of cancer research are consistent with the virological findings of Swingler et al. [50]. According to the latter, the siRNA-dependent silencing of the *MCL1* gene results in a substantial decrease in the level of TRAIL-dependent activation of executive apoptotic enzyme, caspase-3, in the HIV-1-infected macrophages [50].

## 5. MCL-1 in HTLV-1 Infection

Human T-lymphotropic virus type-1 (HTLV-1) belongs to the *Retroviridae* family [57] and the *Deltaretrovirus* genus [58]. Two identical strands of single-stranded positive-sense RNA are located in an enveloped virion approximately 100 nm in diameter [59]. The genome consists of four overlapping open reading frames that encode five regulatory proteins [60]. During infection, the RNA is transcribed by the participation of RT into double-stranded DNA which integrates with the host’s genomic DNA [61,62]. Provirus, which is an integrated virus, serves as a template for producing viral particles [63]. It is estimated that 5 to 10 million people are infected with HTLV-1 and the area of infection includes Africa, southwestern Japan, South America, the Caribbean, and Astralo-Melanesia [64]. HTLV-1 infects CD4+ and CD8+ T cells, B cells, monocytes, and dendritic cells (DCs). Most of the infected individuals remain asymptomatic for life; however, 2–5% of infected people develop adult T-cell leukemia/lymphoma (ATLL) [65,66], HTLV-1-associated myelopathy/tropical spastic paraparesis (HAM/TSP) [67,68], or HTLV-1 uveitis (HU) [69].

HTLV-1 is able to establish a latent infection in T-lymphocytes, which therefore serve as the virus reservoir. Individuals suffering from ATLL are characterized by a high proviral load. The specific feature of the acute phase of this disease is the overrepresentation of HTLV-1-infected CD4 + CD25+ T-lymphocytes in patients’ blood [70,71]. 

The HTLV-1 products affect the ubiquitination pattern of MCL-1 to prevent its degradation. The virus encodes the Tax protein, an agent responsible for the nuclear factor kappa B (NFĸB)-dependent T-cell immortalization. Tax promotes cell survival and proliferation by interplaying with various signaling pathways. It interferes with the molecular events that regulate cell cycle transitions and apoptosis and antagonizes cellular antioncogenic factors [72]. It has been reported that MCL-1 plays a role in the antiapoptotic effect of Tax. The Tax molecule has a TNF receptor-associated factor 6 (TRAF6)-binding motif at its C-terminus, which enables this viral product to directly interact with TRAF6. Meanwhile, TRAF6 is an E3 ubiquitin ligase that catalyzes Lys63-linked polyubiquitination within the target proteins. Tax stimulates the activity of TRAF6 and its migration to mitochondria, promoting this chemical modification in MCL-1. Lys63-linked polyubiquitination confers high stability to MCL-1, thereby increasing the half-life of its molecules and elevating their intracellular amount. After the conjugation with the four ubiquitine polymers, MCL-1 can avoid degradation even under the conditions of genotoxic stress [66,72,73]. Choi et al. [72] used the Cycloheximide (CHX) chase assay to demonstrate that Tax significantly promoted the MCL-1 stability in the Tax-producing mutant of immortalized T lymphocyte cell line, Jurkat [72]. 

On the other hand, HTLV-1 protects MCL-1 from degradative ubiquitination [66], a process involving cullin 1 (CUL1), S-phase kinase-associated protein 1 (Skp1), as well as RING-box protein 1 (RBX-1) that performs the activity of E3 ubiquitine ligase. These three components form the Skp, Cullin, F-box containing (SCF) complex, the structure responsible for marking target proteins for proteasomal degradation [74]. CUL1 binds Skp1 to arrange the molecular scaffold for the recruitment of RBX. Meanwhile, another viral product, HTLV-1 basic leucine zipper factor (HBZ), disrupts the CUL1–Skp1 association, thereby thwarting the assembly of the functional E3 ubiquitin ligase complex [75]. Mukai et al. [75] transfected human embryonic kidney 293T (HEK293T) cells with two vectors encoding HBZ and FLAG-tagged MCL-1. The ectopic expression of the *HBZ* gene meaningly increased the level of MCL-1 compared to the control cells (the ones nonproducing HBZ protein). The team also observed the elevated amount of endogenous MCL-1 in two HTLV-1-positive ATLL cell lines: MT-1 and ATL43T, against the control (i.e., HTLV-1 negative Jurkat cells) [75]. 

These findings suggest that the upregulation of MCL-1 contributes to HTLV-1 latency, the immortalization of infected T-lymphocytes, and the development of ATLL. 

## 6. MCL-1 in HCMV Infection

Human cytomegalovirus (HCMV), also known as human herpes virus 5 (HHV-5), is a member of the *Herpesviridae* family and the *Cytomegalovirus* genus [33]. Its genome has the form of linear, double-stranded DNA [76] and contains at least 165 open reading frames that encode more than 751 transcripts [77]. Its icosahedral capsid is approximately 116 nm in diameter [78]. The infection of HCMV is typically asymptotic and under the control of the immune system in healthy individuals [79]. The infection is lifelong since the virus can establish a state of latency [80]. HCMV infection can cause severe diseases in immunocompromised patients such as cancers, including brain cancer, breast cancer in women, and colorectal cancer [81].

Cytomegaloviruses (CMVs) may hide inside monocytes/macrophages and thus persist in the host body throughout a lifetime. These host cells may perform the function of pathogen reservoirs and favor long-term infection. Moreover, they can play the role of the “bridgeheads” for the possible viral invasion. The latent cell sanctuaries of CMVs enable the virus to spread effectively throughout the body, reach different tissues and organs, and take a “convenient position” for further expansion. Under favorable circumstances, CMVs can reactivate, which may contribute to the development of pathological conditions in immunodeficient hosts, including transplant patients. Monocytes serve as viral vehicles responsible for the hematogenous dissemination of the pathogen. Monocyte-derived macrophages may be either the sites of productive infection or the pathogen reservoirs in the host tissues [82,83,84,85,86].

The effective spread of the pathogen throughout the host organism requires the monocytes to undergo CMV-driven adaptation to the role of viral disseminators. The short circulating lifespan of these cells is a significant challenge for the virus. Moreover, the antiviral signaling increases additionally the apoptotic potential of infected monocytes. Therefore, the pathogen must modify the cell-signaling network to prevent rapid apoptosis and mediate the prosurvival phenotype of the cell [87]. An in vitro study by Chan et al. [87] has demonstrated that the HCMV infection of the human peripheral blood monocytes leads to a temporal increase in the amount of MCL-1. The siRNA-dependent silencing of *MCL1* gene expression enhanced the level of apoptosis in infected monocytes. These results suggest the essential role of MCL-1 in maintaining the monocyte viability upon the initial, nonreplicative phase of viral infection. Further investigation linked the increase in MCL-1 to the activation of the epidermal growth factor receptor (EGFR) and the upregulation of its downstream molecule, phosphatidylinositol 3-kinase (PI3K) [87]. In light of previous research [88], the stimulation of EGFR is the event accompanying the internalization of the virus by the cell [87]. Therefore, neither HCMV replication nor the synthesis of the early viral proteins is required for EGFR/PI3K-dependent upregulation of MCL-1. It explains the antiapoptotic effect of nonproductive HCMV infection [87].

## 7. MCL-1 in Adenovirus Infection

Adenoviruses (AdVs) belong to the *Adenoviridae* family [89] and the *Mastadenovirus* genus [90]. AdVs are nonenveloped, double-stranded DNA viruses. Their icosahedral capsid is approximately 95 nm in diameter [91]. AdVs can infect fish, frogs, reptilians, birds, and mammals [92]. Human adenoviruses (HAdVs) have been grouped into seven species, named A–G. So far, 113 different genotypes and multiple genetic variants have been described [93]. Species B, C, and E can infect the respiratory system, causing pneumonia and acute respiratory distress syndrome. Species B, D, and E are associated with infection of corneal epithelium and conjunctiva causing epidemic keratoconjunctivitis (EKC) or pharyngoconjunctivitis. Species A, F, and G can infect the gastrointestinal tract, causing diarrhea [94].

The AdV serotypes 2 (AdV2) and 5 (AdV5) cause the lysis of infected cells to release viral progeny particles. However, in the meantime, the pathogen maintains the host cell’s viability to fulfil the replication cycle. Apoptosis is an event responsible for the hydrolysis of viral DNA and the limitation of AdV multiplication. Therefore, the virus encodes the BCL-1 homolog, E1B-19K protein, that performs the antiapoptotic function. It antagonizes another AdV product, the E1A molecule. In AdV mutants lacking the *E1B-19K* gene, E1A may initiate the cascade of molecular events leading to apoptosis. E1A plays a role in MCL-1 downregulation that leads to the dissociation of the BAK-MCL-1 heterodimers. Next, the unbound, monomeric BAK molecule binds another proapoptotic representative of the BCL-2 family, BAX, to form heterooligomeric pores in the OMM. The permeabilization of OMM causes the induction of the intrinsic apoptosis pathway, which results in premature cell death, and disrupts AdV replication. In contrast to E1A, the E1B-19K protein performs the mitoprotective function. It counteracts the induction of the intrinsic apoptosis pathway by targeting BAK. By forming complexes with BAK, E1B-19K protein makes it unable to oligomerize with BAX, and thus prevents the permeabilization of the OMM, and the leakage of proapoptotic factors such as cytochrome c into the cytosol. An additional function of E1B-19K is to inhibit Bid, a BCL-2 protein family member responsible for releasing BAK from BAK-MCL-1 complexes. Thus, the wild-type AdV infection causes a decrease in MCL-1 but also counteracts the consequences of its deficiency. E1B-19K can replace the lost MCL-1 in complexes with BAK protein, keeping the latter one neutralized [95,96]. Cuconati et al. [95] have studied the mechanisms responsible for the downregulation of MCL-1 protein in the cells infected with AdV5. The team infected HeLa cervical cancer cells with either wild-type AdV5 or mutant viruses lacking the *E1B 19K* gene. All of them caused the loss of MCL-1. However, the wild-type strain blocked apoptosis. Another mutant, the one deprived of the *E1A* gene, did not affect the level of MCL-1 in HeLa cells, confirming the role of the E1A protein in MCL-1 downregulation. Further analysis revealed that the AdV5-dependent modification of MCL-1 occurred at different levels of *MCL1* gene expression. It resulted from both the decrease in the amount of *MCL1* mRNA and the elevated degradation of the MCL-1 protein. E1A was necessary for the phosphorylation of ataxia-telangiectasia mutant protein (ATM) and the H2A histone family member X (H2AX); the modified forms of these proteins accumulated only in the HeLa cells infected with the viral strains encoded E1A. Meanwhile, the phosphorylated form of ATM and H2AX are the markers of the DNA damage response (DDR). These data suggest that in the context of AdV5 infection, some E1A-dependent DDR-like mechanisms may be involved in MCL-1 degradation [95].

## 8. MCL-1 in EBV Infection

Epstein–Barr virus (EBV) belongs to the *Herpesviridae* family [97] and the *Lymphocryptovirus* genus [98]. Its genome has a form of double-stranded DNA covered by a pseudo-icosahedral nucleocapsid and lipid envelope [97], which has a diameter of 115 nm. Its genome encodes more than 85 genes [99]. EBV is one of the most common human pathogens; 95% of adults worldwide are infected. The infection tends to be asymptomatic and lifelong [100]. EBV latent infection can cause a diverse range of lymphomas, such as Hodgkin lymphoma (HL) [101], post-transplant lymphoproliferative disorders (PTLD) [102], Burkitt lymphoma (BL) [103], diffuse large B cell lymphoma (DLBCL) [104], nasopharyngeal carcinoma (NPC) [105], EBV-positive gastric cancer (EBV-GC) [106], and NK/T cell lymphoma (NKTL) [107].

The EBV genome encodes a set of antiapoptotic proteins enabling the pathogen to establish persistent infection. Three of them, latent membrane protein (LMP)-1, EBV nuclear antigens (EBNA)-2, and -3A, are responsible for the upregulation of MCL-1. The prosurvival viral products are involved in the pathogenesis of EBV-associated malignancies. The synthesis of LMP-1 is observed in different EBV-driven diseases, including DLBCL, HL, NPC, and NKTL. The presence of EBNA-2 and -3A proteins in latently infected cells is characteristic for DLBCL disease. Moreover, to some extent, EBNA-3A may be produced in BL tumor cells. LMP-1 is generally not produced in BL cells, which utilize the Latency I program, thereby restricting the expression pattern to the gene encoding the EBNA-1 protein. However, in in vitro conditions, after a series of subsequent passages, BL cells expand the range of protein production and, inter alia, begin the synthesis of LMP-1 [108]. The antiapoptotic properties of the viral products contribute to EBV-associated oncogenesis, counteracting the death of malignant cells. They interfere with the intracellular signaling network, including MCL-1 as one of its essential nodes [109].

LMP-1 promotes the survival of BL-derived cell lines, upregulating the expression of genes encoding the three antiapoptotic members of the BCL-2 family: MCL-1, BCL-2, and BCL-2-related protein [108]. Wang et al. [110] have determined that transfection with the LMP-1-encoding vector transiently increases the amount of MCL-1 in the DG25 cell line. At 72 h after the induction of the ectopic expression of the *LMP1* gene, the level of MCL-1 returned to a level similar to the initial one, whereas the amount of BCL-2 increased. These findings suggest that the temporary overproduction of MCL-1 might play the cytoprotective role until BCL-2 takes over this function [110].

EBNA-2 also has been found to stimulate the expression of genes encoding BCL-2 family representatives [109]. Kohlhoff et al. [111] used EREB2-5 cells (i.e., the EBV-positive B-lymphocytes able for the estrogen-mediated expression of the *EBNA2* gene) to determine the role of EBNA2 in regulating cell viability. The estrogen treatment elevated the intracellular level of mRNAs encoding several antiapoptotic agents, including MCL-1, BCL-2, BCL-XL, and BCL-2A1 [111].

EBV infection may affect not only the level of MCL-1 but also its intracellular distribution. EBV EBNA-3A promotes the migration of MCL-1 to mitochondria. This localization enables MCL-1 to bind and block proapoptotic BAX. Therefore, EBNA-3A counteracts the induction of the mitochondria-dependent apoptosis pathway [109]. Using the subcellular fractionation technique, Price et al. [112] determined the effect of EBV infection on the distribution of MCL-1 in the lymphoblastoid cell line (LCL). In the cell carrying the wild-type virus, the mitochondrial fraction of MCL-1 protein substantially increased, compared to the control (the cells infected with the EBV mutant lacking the *EBNA3A* gene) [112]. 

In summary, the aforementioned reports suggest that the upregulation of MCL-1 is co-responsible for the latency of EBV infection and the development of EBV-associated diseases, such as BL.

## 9. MCL-1 in KHSV Infection

Kaposi’s sarcoma-associated herpesvirus (KSHV), also known as a human herpesvirus 8 (HHV-8) belongs to the *Herpesviridae* family and the *Rhadinovirus* genus [113]. Its genome has the form of double-stranded DNA [114], and is covered by an icosahedral capsid [115]. The genome has 81 open reading frames that encode up to 90 viral proteins, including structural and metabolic proteins [116,117]. The infection is lifelong and the virus persists in the infected B-cells [118]. KSHV is associated with Kaposi’s sarcoma (KS), pleural effusion lymphoma (PEL), multicentric Castleman disease (MCD), and KSHV-associated inflammatory cytokine syndrome (KICS) [119].

KHSV infection may be associated with primary effusion lymphoma (PEL), a fast-growing B-cell malignancy. The PEL development usually follows the host immune deficiency and culminates in the death of the patients. PEL usually displays the tendency to grow as neoplastic effusions instead of forming solid tumor mass; however, the latter sometimes can also occur. This disease is characterized by very poor survival and high chemoresistance [120]. The KHSV genome consists of the gene encoding latency-associated nuclear antigen (LANA), a product that enables the virus to establish latent infection in the host cells. The antiapoptotic activity of LANA may contribute to the development of KSHV-driven carcinogenesis [121,122]. The molecule of LANA comprises a phosphodegron motif that can interact with the F-box and Trp-Asp (WD) repeat domain-containing 7 (FBW7) protein, rendering it unable to bind MCL-1 [123]. FBW7 is an E3 ubiquitin ligase responsible for the posttranslational modification of a set of target structures, i.e., oncogenic molecules such as cellular Myc (c-Myc) and MCL-1. FBW7-catalyzed ubiquitination leads them to proteasomal degradation [124]. Thus, by blocking FBW7, the KHSV LANA protects MCL-1 from modification that decreases its stability and half-life. It eventually results in the elevated intracellular level of this antiapoptotic agent [123]. Kim et al. [123] compared the amount of MCL-1 in KHSV-infected and KHSV-negative cells. The virus-positive ones included three PEL-derived cell lines: the body cavity-based lymphoma-1 (BCBL-1), BC3, and BC1, the third carrying both KHSV and EBV viruses. BCBL-1, BC3, and BC1 displayed a substantially elevated level of MCL-1 and decreased apoptosis level compared to the KHSV- and EBV-negative BL-derived cell line, BJAB. The siRNA-dependent silencing of the *MCL1* gene expression promotes apoptosis in virus-infected cell lines, whereas BJAB remained unaffected by this intervention. These findings prove the essential role of MCL-1 in maintaining the viability of KHSV-infected cells in the context of PEL [123].

## 10. MCL-1 in Flavivirus Infection

Flaviviruses are a genus of viruses belonging to the *Flaviviridae* family [125]. Their genome has the form of positive-sense single-stranded RNA and is enclosed in an enveloped capsid. The viral RNA consists of a single, long open reading frame (ORF), encoding three structural proteins such as capsid (C), pre-membrane (prM), and envelope (E) protein, and seven non-structural (NS1, NS2A, NS2B, NS3, NS4A, NS4B, NS5) proteins [126]. The NS5 protein is an RNA-dependent RNA polymerase (RdRp) [127]. Flaviviruses pose a serious challenge to medicine. Transmitted by ticks or mosquitoes, they infect up to 400 million people a year. This genus includes viruses such as Japanese encephalitis virus (JEV), Dengue virus (DENV), West Nile virus (WNV), and Zika virus (ZIKV) [128]. JEV is an agent of Japanese encephalitis (JE) [129] and infects up to 68 thousand people each year with mortality around 14–20 thousand [130]. DENV infection, known as dengue fever, leads to the hospitalization of 500 thousand people and 20 thousand deaths out of 390 million infections a year [131]. WNV infection includes symptoms such as fever, headache, body aches, nausea, vomiting, diarrhea, swollen lymph glands, rash, and muscle and joint pain. Less than 1% of infected individuals suffer from meningo-encephalitis, which can cause death [132]. ZIKV causes Zika fever, which usually progresses as a mild infection. About 25% of those infected experience symptoms such as fever, rash, joint and muscle pain, and weakness [133].

A study by Suzuki et al. [134] on the Huh7 HCC cell line has shown that infections with flaviviruses such as JEV, DENV, and ZIKV decrease the intracellular level of MCL-1. The use of a radiolabeled amino acid, i.e., [35S]-methionine (intended to be embedded into the target structure), allowed the team to assess the impact of JEV infection on the level of newly synthesized MCL-1 protein. Over three days, the incorporation rate decreased as the infection progressed. Meanwhile, the level of mRNA remained unaffected, suggesting that in Huh7 cells, JEV infection downregulates the *MCL1* gene expression at the level of translation. Further experiments revealed the divergence in JEV-dependent MCL-1 regulation among different cell lines, depending on their susceptibility to the virus propagation. IGROV-1 (human ovarian carcinoma cells), H522 (lung adenocarcinoma cells), and SF-268 (human glioblastoma cells) responded to JEV infection, with a substantial fall in MCL-1 synthesis. Meanwhile, OVCAR5 (human ovarian cancer cells) and HT29 (human colorectal adenocarcinoma cells) displayed a stable level of MCL-1. The first set of cell lines was characterized by a high susceptibility to JEV propagation and the second group with a low one. The analysis of the MCL-1 level in either the JEV, ZIKV, or DENV-infected human lymphoma cells, U937, showed the correlation between the MCL-1 downregulation and the effectiveness of virus propagation (manifested in the viral titer). ZIKV infection caused the highest fall in the MCL-1 level, and the viral propagation was the most efficient. On the contrary, the U937 cells displayed the weakest sensitivity to DENV infection, and the lowest viral titer was consistent with the stable expression of the *MCL1* gene. These findings indicate that the flavivirus-dependent suppression of the synthesis of MCL-1 protein determines the pathogen propagation. Interestingly, the JEV-, ZIKV- or DENV-infected Huh7 cells maintained a stable level of another antiapoptotic member of the BCL-2 family, the BCL-XL. This protein was a key factor determining the cell viability during the infection. BCL-XL prevented premature apoptosis, counter to the deficiency in MCL-1. Thus, by affecting the MCL-1, but not the BCL-XL level, flaviviruses harmonize the action of the apoptosis regulation network with their replication cycles, and the appropriate timing allows the pathogen to propagate effectively [134].

## 11. MCL-1 in IAV Infection

Influenza A viruses (IAVs) are members of the *Orthomyxoviridae* family [135]. Its genome has the form of negative-sense single-stranded viral RNA (vRNA) [136] and is divided into eight segments, encoding few proteins; polymerase proteins (PB1, PB2, and PA), surface glycoproteins (hemagglutinin HA and neuraminidase NA), matrix proteins (M1 and M2), nucleoprotein (NP), and non-structural proteins (NS1 and NS2) [137]. Each segment of vRNA consists of one or two open reading frames (ORFs) [138]. The enveloped virion has two forms: spheres or elongated forms, approximately 100 nm in diameter [139]. The IAV serotypes are determined by variants of HA and NA [140]. So far, 18 HA (H1-18) and 11 NA (N1-11) subtypes have been described [141]. IAVs constitute a global health problem due to annual epidemics that result in millions of human infections. Viruses constantly acquire new changes in the form of point mutations or antigenic changes [142]. IAVs can infect animals such as birds [143], horses [144], swine [145], and dogs [146] and pose a risk of zoonotic infections [141].

The efficacy of IAV propagation depends on the host cell viability. Thus, the intracellular antiapoptotic signaling creates circumstances for successful IAV replication [147]. To discover the mechanism responsible for cell survival during the viral infection, Weiss et al. [148] performed siRNA-dependent silencing of the *MCL1* gene in Vero cells. The MCL-1 deficiency elevates the apoptotic potential of the IAV-infected cells to the level resulting in the virus-driven activation of caspase-3 and a consequent substantial decrease in viral titers. These findings suggest that MCL-1 may counteract apoptosis to the IAV’s advantage. However, whether IAV infection affects the amount of MCL-1 protein in Vero cells or benefits only from the constitutive MCL-1 synthesis has not been determined [148]. Instead of this, Denisova et al. [149] observed a temporal increase in MCL-1 within the IBV-infected retinal pigment epithelial (RTE) cell line [149].

On the other hand, IAV has been found to downregulate the level of MCL-1 in neutrophils. These immune cells are responsible for the removal of IAV from the host lungs, and thus the virus reaps the benefits from their death. Research in mice [150] has shown that IAV subtype H1N1 infection elevated the apoptosis rate in neutrophils within the lung, whereas the proinflammatory cytokine, IL-6, counteracted this effect, promoting cell survival. Further analysis linked the above-mentioned observations to the regulation of the MCL-1 level. IAV infection led to the substantial loss of MCL-1, and the IL-6 treatment reversed this change, reverting MCL-1 to the normal level [150]. IL-6 is a chemical messenger that promotes the survival of neutrophils and thus determines the effectiveness of the innate antiviral immune response in the infected lung. Upregulating MCL-1 in neutrophils, IL-6 allows them to eliminate the pathogen, thereby preventing the efficient viral propagation, the consequent lung damage, and the fatal outcome of the infection [150,151,152].

## 12. MCL-1 in Coronavirus Infection

Coronaviruses (CoVs) are a group of viruses belonging to the *Coronaviridae* family and the *Orthocoronavirinae* subfamily [153]. Their genome is a positive-sense single-stranded RNA and is enclosed in an enveloped, helically symmetric nucleocapsid [154]. The CoVs’ genome contains at least six open reading frames, encoding structural proteins such as the spike (S), membrane (M), envelope (E), and nucleocapsid (N) protein [155]. Three highly pathogenic human coronaviruses (hCoVs): severe acute respiratory syndrome coronaviruses (SARS-CoV), SARS-CoV-2, and Middle East respiratory syndrome coronavirus (MERS-CoV), caused three worldwide epidemics. SARS-CoV-2 is an etiological agent of coronavirus disease 2019 (COVID-19) [156]. On 11 March 2020, the World Health Organization declared COVID-19 a global pandemic [157]. The end of the pandemic was announced on 5 May 2023 [158]. During this time, the virus has infected more than 670 million people and contributed to nearly 7 million deaths, becoming one of the biggest challenges of modern medicine [159].

CoV infection affects the apoptotic potential of the host cells. Their viability influences the efficacy of pathogen propagation and consequent virus titer. Moreover, studies on the mechanisms determining the death/survival balance of the CoV-carrying cells provide a possible explanation of the frequent, asymptomatic course of the infection. Meanwhile, the lack of clinical manifestations hinders the proper diagnosis and facilitates pathogen dissemination through the population. Therefore, the antiapoptotic molecular events, an essential part of the host–virus interplay, may implicate far-reaching epidemiologic consequences [160,161].

Zhong et al. [161] have studied avian coronavirus infectious bronchitis virus (IBV) infection as a model for the research on the molecular interaction between CoV and the cell. In vivo and in vitro experiments have revealed the key role of the two antagonistic molecules, MCL-1 and BAK, in determining the viability of the infected cell. In response to IBV infection, both proteins were upregulated. The increase in the level of *MCL1* and *BAK* mRNA was observed in IBV-infected African green monkey kidney epithelial cells, Vero, and in three other virus-carrying cell lines, i.e., chicken fibroblasts—DF1, lung cancer cells—H1299, and Huh7. The transcriptional upregulation resulted in increased protein production. The tested cell lines responded to IBV infection by elevating the amount of MCL-1 and BAK. Moreover, in vitro studies were enriched with the in vivo experiments. The IBV infection increased the levels of *MCL1* and *BAK* mRNA in embryonated chicken eggs [161]. 

Further analysis linked the IBV-dependent upregulation of MCL-1 to the disruption of endoplasmic reticulum (ER) homeostasis. The increase in the MCL-1 level required the activity of growth arrest- and DNA damage-inducible gene 153 (GADD153), a molecule that plays a role in ER stress signaling. Moreover, two molecular events downstream of ER stress determined the expression of the *MCL-1* gene in IBV-infected cells, i.e., the activation of the PI3K/protein kinase B (PKB) pathway as well as the rat sarcoma virus proto-oncogene (Ras)/rapidly accelerated fibrosarcoma proto-oncogene (Raf)/mitogen-activated protein (MAP) kinase/ERK kinase (MEK)/extracellular-signal-regulated kinase (ERK) pathway. The use of MEK-1 or PI3K inhibitor, or the siRNA-induced silencing of GADD153 expression led to a substantial downregulation of MCL-1 synthesis in both Vero and H1299 cells. Interestingly, the overproduction of MCL-1 in mammalian cells transfected with the *MCL1* gene-carrying vector reduced the release of IBV protein/particles to the culture medium, compared to the control. Meanwhile, the siRNA-induced knockdown of *MCL1* gene expression caused the opposite effect. These findings suggest that the elevation of MCL-1 in IBV-infected cells might be the manifestation of the host immune defense mechanism that is targeted at the reduction in virus propagation. By delaying apoptosis, MCL-1 might hinder the release of progeny virions and counteract effective viral dissemination [161].

However, the study by Pan et al. [160] suggested the opposite role of MCL-1 in cell infection with another representative of the *Coronaviridae* family, SARS-CoV-2. The team demonstrated that one of the viral products, N protein, upregulated the MCL-1 level and, inhibiting apoptosis, promoted the pathogen’s propagation. To obtain a system for studying the SARS-CoV-2–cell interactions, the human colorectal adenocarcinoma cell line producing the N protein, Caco2-N, was infected with the constructs lacking the *N* gene, i.e., SARS-CoV-2 tetracistronic transcription- and replication-competent virus-like particles (SARS-CoV-2-trVLP). The treatment of the Caco2-N cells with the potent inhibitor of MCL-1, S63845, caused a substantial increase in the number of viral copies, which was a measure of replication efficiency. It suggested that in contrast to IBV, SARS-CoV-2 benefits from the antiapoptotic activity of MCL-1, and the upregulation of this protein is a proviral event. Pan et al. [Pan] also revealed the mechanism responsible for the SARS-CoV-2-mediated increase in MCL-1. Unlike in the case of IBV infection, the *MCL1* gene expression underwent posttranslational regulation, and the *MCL1* mRNA remained unaffected. The SARS-CoV-2 N protein bound Pro-, Glu-, Ser-, and Thr-rich (PEST)-like domain of MCL-1 and recruited ubiquitin-specific peptidase 15 (USP15). Cutting the ubiquitin chain off of Lys63, USP15 increased the stability of MCL-1. The antiapoptotic properties of the N protein were confirmed by in vivo experiments. Mice infected with adeno-associated viruses carrying the *N* gene displayed an elevated level of apoptosis within the lung tissue, compared to the control (the animals nonproducing N protein) [160].

In summary, MCL-1 plays either a pro- or antiviral role in CoV infections. The upregulation of MCL-1 is the event that reduces the dissemination and propagation of IBV. However, it also promotes SARS-CoV-2 infection.

## 13. MCL-1 in VSV Infection

Vesicular stomatitis virus (VSV) represents the *Vesiculovirus* genus of the *Rhabdoviridae* family [162]. The term „vesicular stomatitis virus” comprises a certain number of related viruses that belong to one serogroup, with the two most prevalent being New Jersey and Indiana serotypes [163]. The genome of VSV consists of nonsegmented, negative-sense, single-stranded RNA (ssRNA) approximately 11 kb long, enclosed in a bullet-shaped virion, and encodes five proteins: nucleoprotein (N), phosphoprotein (P), matrix protein (M), glycoprotein (G), and polymerase (L) [164]. During infection, the expression of the host’s genes is turned off, and the virus takes over the entire metabolic potential of the cell. VSV infects domestic animals and causes economically relevant disease, the symptoms of which resemble foot-and-mouth disease. The severe ulceration and/or vesiculation of the muzzle, tongue, and oral tissues as well as udder, prepuce, and feet, translate into a significant loss of productivity [163]. In addition to cattle, horses, pigs, and arthropods, which are vectors for the virus, VSV can also easily infect humans and cause a temporarily debilitating influenza-like disease [163,165], or, due to its neurovirulence, cause severe encephalitis [166]. VSV is widely used as a platform for oncolytic vectors or vaccine delivery agent, e.g., against SARS-CoV-2 and influenza A virus infections [167,168].

VSV infection leads to mitochondria-dependent apoptosis. The molecular scenario ending in the death of infected cells includes the downregulation of the level of MCL-1 and the inactivation of BCL-XL. A study by Pearce and Lyles [169] on HeLa cells revealed the ability of the VSV to decrease the expression of the *MLC1* gene and linked this phenomenon to BAK activation and apoptosis. The viral product responsible for MCL-1 downregulation was the VSV M protein, a powerful inhibitor of the host gene expression. The MCL-1 protein is characterized by a very short half-life. The decrease in the de novo synthesis of MCL-1, together with the proteasomal degradation of existing MCL-1 molecules, resulted in the substantial decrease of the MCL-1 level. Meanwhile, the amount of BCL-XL remained unaffected. However, the use of siRNA to silence the expression of the *MCL1* and/or *BCLX-L/S* gene showed that the VSV-driven induction of apoptosis requires the deficiency in the activities of both proteins—MCL-1 and BCL-XL—and thus the posttranslational inactivation of the latter one must have occurred [169]. 

VSV can induce apoptosis in many different cell types. Due to its efficiency in cell killing, the proapoptotic virus is considered a potential therapeutic agent in oncological medicine. Various scientific reports discuss the place of VSV and its oncolytic features in the future landscape of cancer virotherapy [170,171,172,173]. The study by Schache et al. [174] determined the key role of VSV-dependent MCL-1 downregulation in the treatment of AML [174,175].

## 14. MCL-1 Protein in Antiviral Therapies and Treatments of Virus-Associated Diseases: Current Knowledge and Future Perspectives

As previously mentioned, the antiapoptotic activity and mitoprotective function of MCL-1 determine the viability of the host cells, either the uninfected or the virus-infected ones. As an important node of the cell death signaling network, the MCL-1 protein may contribute to the maintenance of the viral reservoir. It can also promote the progression of virus-associated cancers by desensitizing the malignant cells to apoptotic stimuli, including chemotherapeutic agents [21,44,50,72,87]. Therefore, the direct or indirect targeting of MCL-1 seems to be a promising strategy for either viral infection treatment or the therapy of virus-derived malignancies. On the other hand, the virus-dependent loss of the MCL-1 protein may lead to the aberrant, intensified apoptosis of the host cells and the consequent damage of the tissues. Thus, the agents preventing this scenario also are in demand in medicine [24]. Table 1 presents the antiviral compounds affecting MCL-1 and the mechanisms of their actions in the fight against different viruses.

### 14.1. BH3 Mimetics—The Direct Inhibitors of the MCL-1 Protein Activity

#### 14.1.1. S63845

Several studies have been devoted to the direct targeting of MCL-1 in the fight against viral infections. S63845 is a small-molecule polycyclic compound designed to inhibit MCL-1. S63845 can bind MCL-1 with high specificity and affinity, blocking the hydrophobic BH3-binding pocket, a structure necessary for the interaction between MCL-1 and other BCL-2 family members, including BAK. S63845 has been reported to possess antioncogenic activities. Both in vitro and in vivo research has lent support to the idea of using S63845 in the treatment of MCL-1-promoted cancers [176,177,178]. Its potential in antiviral therapies has also been discussed [179].

Inde et al. [179] performed an in vitro experiment to evaluate the effectiveness of S63845 in fighting SARS-CoV-2 infection. The team did not observe the individual impact of the tested compound on the rate of apoptosis in the infected Vero cells. However, S63845 turned out to synergize with the two other BH3 mimetic drugs, either ABT-199 or ABT-737. The first one inhibits BCL-2, and the second one targets BCL-2, BCL-XL, and BCL-w. Supporting ABT-199 or ABT-737 with S63845 led to a noticeable acceleration of apoptosis [179].

An in vitro study by Sejic et al. [180] was devoted to the possible effectiveness of BH3 mimic agents in the treatment of EBV-associated diseases [180]. As described before, EBV infection may be associated with various malignancies, including NKTL [109]. The most prevalent subtype of NKTL is the extranodal NKLT (ENKTL). The pathogenesis of this malignancy requires the proinflammatory cytokine, IL-2. By activating its specific receptor, this chemokine stimulates the development of the cancer and renders it resistant to chemotherapy [181]. Sejic et al. [180] demonstrated that the S63845 treatment enhanced apoptosis in the ENKTL cell line, SNT15. The withdrawal of IL-2 intensified the proapoptotic effect of the MCL-1 inhibitor on the SNT15 cells. Moreover, the lack of IL-2 stimulation sensitized another ENKTL cell line, MEC04, to S63845. The team also demonstrated the synergical relationship between the different BH3 mimetic agents. The results obtained from cell culture models suggested the substantial benefits of combining the S63845 with the drug targeting BCL-XL, A-1331852. In the two IL-2-deprived ENKTL cell lines, SNT15 and SNK6, the combination treatment with S63845 and A-1331852 was more effective than using A-1331852 individually [180].

S63845 was also tested against KHSV [182]. As previously mentioned, this virus is associated with PEL, an aggressive B-cell cancer. This disease is characterized by very poor survival accompanying the high resistance to conventional chemotherapeutic treatments [120], which has motivated scientists to search for more effective antioncogenic agents [182]. Dunham et al. [182] revealed that S63845 induced apoptosis in KHSV-positive as well as KHSV- and EBV-positive cells, BC3 and BC1, respectively. Interestingly, these cell lines turned out to be resistant to inhibitors targeting either BCL-2 or BCL-XL, indicating the main role of MCL-1 in maintaining the viability of virus-infected PEL cells [182]. 

The antiviral potential of S63845 was also confirmed by research in vivo. Arandjelovic et al. [183] have studied the effectiveness of the BH3 mimetic drugs in treating HIV infection in the humanized mouse model. S63845 synergized with the BCL-2 inhibitor, venetoclax, to extend the delay in the HIV rebound after the discontinuation of antiretroviral therapy [183].

#### 14.1.2. Obatoclax

Another direct inhibitor of the MCL-1 protein is obatoclax. Compared to S63845, it shows a wider range of target structures: it can block the activity of MCL-1, BCL-2, and BCL-XL [184]. Most probably, obatoclax targets the hydrophobic BH3-binding groove of MCL-1. Thus, the inhibitor effectively prevents the formation of the MCL-1–BAK complex, thereby contributing to OMM permeabilization and consequent apoptosis [185]. The anticancer potential of obatoclax has been assessed in several early-stage clinical trials. The use of MCL-1 inhibitor showed some therapeutic effects in hematologic cancers such as AML, chronic lymphocytic leukemia (CLL), and myelofibrosis [186]. 

Meanwhile, Denisova et al. [149] used PBMC-derived macrophages and RTE cells for the preclinical evaluation of the anti-IAV activity of obatoclax. The drug disturbed the IAV propagation, which was manifested as reduced viral titers. The team determined the role of MCL-1 inhibition in a fight with the pathogen. The RTE cells were infected with a green fluorescent protein (GFP)-encoding IAV strain and underwent partial siRNA-dependent silencing of the *MCL1* gene expression. The loss of MCL-1 resulted in a moderate elevation of the apoptosis rate in infected cells and the impairment of viral propagation (measured by GFP level) [149].

#### 14.1.3. AT-101 and A-1210477

Two other BH3 mimetics, AT-101 and A-1210477, have been found to influence the KHSV-infected cells through blocking the activity of MCL-1 [123]. A-1210477 displays specificity towards MCL-1, whereas AT-101 has a wider range of targets, including MCL-1, BCL-2, BCL-XL, and BCL-w [123,187,188,189]. A study by Kim et al. [123] showed that both inhibitors promoted the apoptosis in virus-infected BCBL-1, BC3, and BC1 cells, while the BJAB cell line, the only one carrying neither KHSV nor EBV, remained merely under slow, inconsiderable influence. Next, ex vivo experiments revealed the reducing effect of AT-101 on the ability of the KHSV-positive cells (BCBL-1) to form colonies. These findings suggested the essential role of MCL-1 in tumorigenesis in the course of KHSV-associated PEL. Moreover, the team used the xenograft mouse model to examine the effectiveness of AT-101 in vivo. BBCL-1 cells as well as AT-101 or dimethyl sulfoxide (DMSO) were injected into mice via the intraperitoneal (IP) route. The AT-101-receiving individuals displayed the substantially milder symptoms of the disease, compared to the control. The AT-101 treatment reduced the volume of ascites and prevented splenomegaly [123].

### 14.2. Imatinib

The study by Swingler et al. [50] delivered on promises to enrich the therapeutic arsenal against HIV infection with imatinib [50]. This is a drug targeting the M-CSF receptor (M-CSFR), also known as McDonough feline sarcoma viral oncogene (FMS). FMS is a class III receptor tyrosine kinase (RTK) characterized by specificity towards M-CFS, and thus has an important role in monocyte-to-macrophage differentiation [50,190,191,192]. As described before, the HIV-induced M-CSF upregulates the expression of the MCL1 and BFL1 genes in HIV-infected macrophages, conferring the resistance of these viral sanctuaries to TRAIL-dependent killing [50]. Swingler et al. [50] have tested the effectiveness of imatinib in the sensitization of HIV-positive cells to TRAIL. In primary human macrophages, the drug prevented the ligand-dependent tyrosine autophosphorylation of M-CSFR, a chemical modification responsible for the activation of this receptor. Moreover, cytometry analysis showed that the virus-infected cells responded to imatinib with a substantial increase in the TRAILR1 expression. This effect together with the M-CSFR inhibition elevated the level of TRAIL-induced apoptosis in the HIV-infected macrophages, compared to the virus-positive cells that were not treated with the drug. As expected, the use of imatinib caused a significant drop in the MCL-1 protein level. Thus, this BCL-2 family member turned out to be the indirect target of a promising antiviral agent [50].

### 14.3. Sorafenib

Another therapeutic compound affecting the level of MCL-1 protein is sorafenib, an anti-cancer multikinase inhibitor. It can block the activity of some receptor tyrosine kinases as well as serine/threonine kinases to mitigate neoangiogenesis in the context of miscellaneous malignancies, including HCC and renal cell carcinoma (RCC) as well as colon, pancreas, and breast carcinoma. Sorafenib is known to inhibit the Ras/Raf/MEK/ERK pathway. Meanwhile, Ras/Raf/MEK/ERK is a signaling cascade that, upon pathological activation, promotes oncogenesis, favoring the proliferation of the tumor cells [193,194,195,196,197]. 

Research has revealed the ability of this drug to decrease the MCL-1 level in malignant cells of human colorectal adenocarcinoma, renal cancer, human breast adenocarcinoma, cholangiocarcinoma, acute T-cell leukemia, chronic myeloid leukemia (CML), and CLL [198]. Interestingly, sorafenib does not affect the quantity of MCL1 mRNA but acts at the translational and post-translational level. The drug-induced mechanism of the MCL-1 protein elimination is the proteasomal proteolysis of this BCL-2 family member. Moreover, it has been reported that sorafenib can counteract the phosphorylation of eukaryotic initiation factor 4E (eI4E). The drug suppresses the chemical modification of eI4E via the Ras/Raf/MEK/ERK-independent pathway, possibly through blocking the activity of (MAPK)-interacting serine/threonine kinase (MNK1). Affecting the eI4E, sorafenib disturbs the translation of the MCL1 gene. As the MCL-1 is the mitoprotective factor, the consequence of its downregulation is the considerable increase in the cell sensitivity to the mitochondria-mediated apoptotic pathway [198,199,200]. 

The efficiency of sorafenib in the treatment of both HBV- and HCV-associated HCC has been tested in clinical research. Recently, Lee et al. [201] used this medicine on 444 patients suffering from HCC and infected with HBV or HCV, and observed an OS of approximately 8.8 months, regardless of the virus species. However, in most of the HBV-positive patients, the treatment with sorafenib was preceded with antiviral therapy based on nucleotide analogs (NAs), whereas only a small percentage of the HCV-infected individuals had previously undergone the anti-HCV medical procedures (i.e., treatment with the use of pegylated interferon (PEG-INF) in combination with ribavirin). Thus, these data suggest that in patients suffering from HBV-associated HCC, the therapy with sorafenib should be supported with the administration of antiviral agents such as NA [201].

In preclinical research, Mao et al. [21] determined the effectiveness of sorafenib in the treatment of HBV-associated HCC. HepG2.2.15 cells, a model system for research on HBV-infected malignant hepatocytes, displayed an elevated resistance to sorafenib compared to HBV-nonexpressing HCC cells, HepG2. These findings suggested that the HBV-dependent decrease in miR-193b and consequent increase in MCL-1 enhance the cell viability and promote its survival upon the treatment with the proapoptotic drug. However, the combination of sorafenib with miR-193b mimics improved substantially the effectiveness of the treatment. The transient transfection of HepG2.2.15 cells with miR-193b RNA-like fragments impaired the cell resistance to the tested medicine. miR-193b mimics increased the apoptosis rate after 72 h exposure to sorafenib, compared to the effect of this drug alone. The half-maximal inhibitory concentration (IC50) of sorafenib was relevantly lower in transfectants than in miR-193b-mimic-nontransfected cells. These findings open up a perspective on the use of MCL-1-decreasing agents in the treatment of HBV-associated HCC [21].

### 14.4. Antioxidants

The elevated MCL-1 level desensitizes the HBV-positive HCC cells to apoptotic stimuli, favoring cancer maintenance and development. However, upon the antioncogenic treatment of the liver with the chemotherapeutic drug cisplatin, the HBx-dependent downregulation of MCL-1 may also be the pathogenic event. In this context, the deficiency in MCL-1 can promote excessive apoptosis and the consequent hepatocellular injury. Thus, the MCL-1-upregulating strategies seem to be the promising solution for protection against liver damage in cisplatin therapy. The antioxidants can counteract the loss of MCL-1 protein, cell apoptosis, and consequent damage to the liver. As described before, NAC and GSH saved MCL-1 in HepG2-HBx and HepG2.2.15 cells [24]. Research in vivo by Hu et al. [24] determined the influence of butylated hydroxyanisole (BHA) on the MCL-1 level in the liver of HBx-positive mice transfectants treated with cisplatin. The intake of food with added BHA diminished the decline in the amount of MCL-1 and limited apoptosis. These findings provide the reasons for combining cisplatin with antioxidants during the HBV-associated HCC treatment [24]. Another rationale for preventing the development of oxidative stress in HBV-infected hepatocytes is cisplatin-induced autophagy that favors viral replication [202,203]. So far, miscellaneous natural antioxidative extracts and compounds have been tested for the ability to reduce the hepatotoxicity of cisplatin in the treatment of HCC. The investigations are at the level of preclinical studies, and the effectiveness of these antioxidants in limiting cisplatin hepatocytotoxicity in HBV-positive HCC patients remains unknown. The need for successful treatment strategies that would be free from side effects suggests the directions for future investigations [204,205].

### 14.5. Resveratrol

Resveratrol is a stilbenoid, a natural plant-derived polyphenol compound [206,207]. An in vitro study by Suzuki et al. [208] delivered on promises to consider resveratrol in a treatment of ATLL, an HTLV-1-driven malignancy. As conventional chemotherapy remains ineffective in ATLL treatment, novel therapeutic strategies are necessary. The team utilized the HTLV-positive ATLL cell lines, MT-2 and HUT-102, to determine preliminarily the antiviral potential of resveratrol and characterize the mechanism of the action of this agent. The tested drug suppressed the phosphorylation of two amino acid residues, Tyr705 and Ser727, within the signal transducer and activator of transcription (STAT)3 molecule [208]. The activity of STAT3 had been previously linked to the apoptotic and proliferative potentials of the cells. The active, phosphorylated form of STAT3 was found to cause an increase in a set of antiapoptotic agents, including the inhibitors of apoptosis protein-2 (c-IAP2) and MCL-1 [209]. Indeed, Suzuki et al. [208] demonstrated that resveratrol treatment resulted in the dose-dependent down-regulation of c-IAP2 and MCL-1 levels in both HTLV-positive ATLL cell lines. The exposure to S3I-201, a STAT3 inhibitor, inhibited apoptosis and diminished the proliferative potential of the cells. The immunoblot analysis of the S3I-201-treated HUT-102 cells revealed a decrease in both the c-IAP and MCL-1 levels, suggesting that resveratrol antagonized the prosurvival effect of HTLV-1 infection by affecting the phosphorylation state of STAT3 [208].

### 14.6. IL-24

As mentioned before, the MCL-1 protein promotes viral propagation by enhancing the viability of infected cells [147]. Weiss et al. [148] lent support to consider the enrichment of a medical anti-IAV arsenal with IL-24. It was able to promote the Toll-like receptor (TLR)3-dependent apoptosis in IAV-infected Vero cells [148]. IL-24 is a cytokine characterized by both antiviral and anticancer properties. An in vitro study on prostate carcinoma cells demonstrated that IL-24 could cause ER stress, resulting in a substantial decrease in the MCL1 gene expression at the level of translation, and leading to apoptosis [147,210]. Meanwhile, Weiss et al. [148] observed the downregulation of the MCL-1 level in IAV-infected Vero cells, suggesting that the loss of this protein is responsible for the proapoptotic and antiviral effect of IL-24 [148].

## 15. Conclusions

The apoptosis regulation network is a complex system of miscellaneous signaling molecules determining cell fate under various conditions. Viral infections affect the host cell viability. They either accelerate or delay apoptosis, depending on the virus species, cell type, and biochemical, physiological, and immunological context of the pathogen–host interaction. The substantial change in the apoptotic potential of infected cells may be the effect of the antiviral immune response or fit the virus strategies for effective replication and dissemination. A large set of viral species influences the intracellular level of the MCL-1 protein, thereby modifying the apoptotic potential of infected cells. So far, several reports have demonstrated the virus-dependent upregulation of MCL-1. The increase in intracellular level of this protein has been observed in HIV, HTLV-1, HCMV, EBV, KHSV, coronaviruses (IBV and SARS-CoV-2), and VSV infections. HBV and IAV upregulate or decrease the amount of MCL-1, depending on the pathophysiological conditions (HBV) or the type of infected cells (IAV). The infections with HCV, flaviviruses (JEV, DENV, and ZIKV), AdV, and VSV cause the loss of MCL-1. The change in the MCL-1 level plays an important role in pathogen replication and dissemination, impairs the host immune response, and contributes to the persistence of infection. Moreover, the deregulation of MCL-1 promotes the development of viral-associated diseases and pathological processes, including liver damage, HCC, ATLL, BL, and PEL. The upregulation of MCL-1 is an event favoring the latency of HIV and HCMV in monocytes/macrophages, HTLV-1 in T-lymphocytes, and EBV and KHSV in B-cells. The HCV core protein can bind and block MCL-1. Research also suggests that chronic HCV infection may indirectly lower the MCL-1/BIM ratio to decrease the apoptotic potential and reactivity of the antigen-stimulated CTLs. Similarly, IAV downregulates MCL-1 in lung neutrophils to avoid the removal of the pathogen from the infected lungs. Flaviviruses and adenoviruses reorchestrate the apoptosis regulation network to adapt it to the dynamics of the replication cycle, and the downregulation of MCL-1 is an event ensuring the appropriate timing for effective propagation. MCL-1 plays different roles in infections with CoVs. The MCL-1 upregulation in IBV-infected cells is suspected to be the host immune defense mechanism that thwarts the dissemination and reduces the propagation of the pathogen. On the contrary, SARS-CoV-2 benefits from the elevation of MCL-1, as the high intracellular level of this protein is associated with increased viral titer. 

Viral infections and virus-driven diseases remain a serious challenge to the molecular sciences and modern medicine. Several studies have delivered on promises to consider MCL-1 as a target structure in anticancer and antiviral therapies. Potential therapeutic agents include direct inhibitors of MCL-1, the BH3 mimetics, i.e., S63845, obatoclax, AT-101, and A-1210477. They interact with the MCL-1 molecule to suppress its antiapoptotic effect. The mechanism of the direct inhibition of MCL-1 is to block the hydrophobic BH3-binding pocket, a structure responsible for the interaction between MCL-1 and other BCL-2 family representatives. So far, preclinical studies have determined the antiviral activity of BH3 mimetics in SARS-CoV-2, EBV, KHSV, and IAV infections. Research has also indicated the compounds that indirectly affect the level of MCL-1, including imatinib, sorafenib, antioxidants (e.g., BHA), resveratrol, and IL-24. Imatinib targets M-CSFR and causes the indirect downregulation of MCL-1 in HIV-infected macrophages, thereby sensitizing them to TRAIL-dependent killing. In light of in vitro studies, sorafenib stimulates the proteolytic degradation of MCL-1 in HBV-infected HCC cells, thus elevating their apoptotic potential. Importantly, the effectiveness of this drug was demonstrated in clinical studies devoted to HBV- and HCV-associated HCC treatment. Antioxidants antagonize the HBx-dependent downregulation of MCL-1 in HBV-associated HCC cells treated with cisplatin. It cannot be excluded that combining cisplatin with antioxidant compounds might prevent the excessive apoptosis of hepatocytes and limit liver injury. Resveratrol affects the phosphorylation status of STAT3 and indirectly downregulates the level of MCL-1 in HTLV-positive ATLL cells, abolishing the prosurvival effect of viral infection. Meanwhile, IL-24 has been found to downregulate the translation of *MCL1* mRNA and favor the TLR3-dependent apoptosis in some IAV-infected cells.

The discussed research points out that the development of MCL1 inhibitors implicated in pathologic cell survival is justified. Furthermore, understanding the biology of MCL-1 and other anti-apoptotic BCL-2 family proteins during viral infections may help to explore the therapeutic rationale for MCL-1 inhibitors as a novel treatment for human cancers and other diseases. So far, the antiviral activity of MCL-1-targeting agents has been examined mainly in cell models. Thus, further extensive investigations are needed to confirm the promising initial findings.

## Figures and Tables

**Figure 1 ijms-25-01138-f001:**
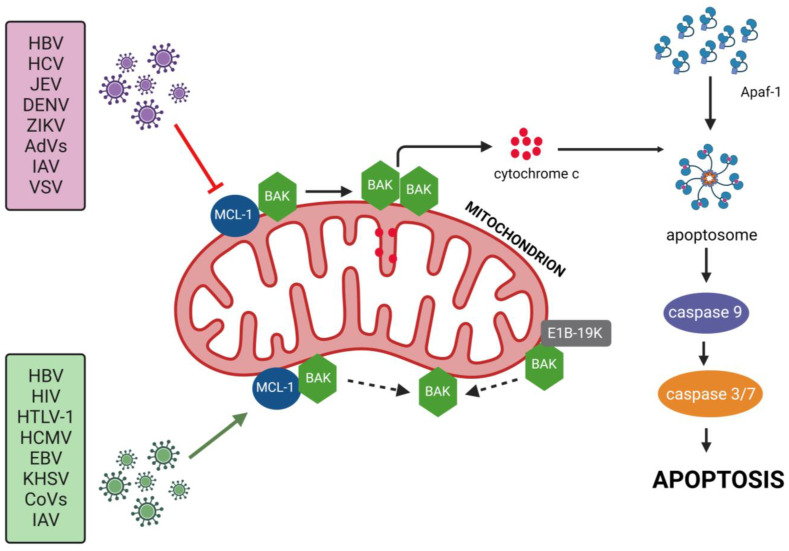
The influence of viral infections on the Mcl-1 protein. The colors of the arrows represent the effect of different pathogens on the intracellular amount or activity of Mcl-1 (green: upregulation; red: downregulation). The continuity of the black arrows symbolizes the translocation of the given molecule(s) to the target structure(s), whereas the discontinuity means the inhibition of this event. The viruses affecting the level of Mcl-1 include hepatitis B virus (HBV), hepatitis C virus (HCV), human immunodeficiency virus (HIV), human T-lymphotropic virus type 1 (HTLV-1), human cytomegalovirus (HCMV), adenoviruses (AdVs), Epstein–Barr virus (EBV), Kaposi’s sarcoma-associated herpesvirus (KHSV), Japanese encephalitis virus (JEV), Dengue virus (DENV), Zika virus (ZIKV), influenza type A virus (IAV), infectious bronchitis virus (IBV), severe acute respiratory syndrome coronavirus 2 (SARS-CoV-2), and vesicular stomatitis virus (VSV). The cellular proteins involved in regulation, initiation, and execution of mitochondria-dependent apoptosis are myeloid leukemia cell differentiation protein 1 (MCL-1), BCL-2 homologous antagonist/killer (BAK), apoptotic protease activating factor 1 (APAF-1), cytochrome c, and caspases (-9, -3, and -7). E1B19K is the AdV protein that can replace Mcl-1 in complex with Bak to delay apoptosis in AdV-infected cells.

**Table 1 ijms-25-01138-t001:** The antiviral agents affecting the activity or the level of MCL-1 in infected cells.

Agent(s)	Effect on MCL-1	The Perpetrator(s) of Treated Infection(s)/Disease(s)
S63845	direct inhibition	SARS-CoV-2 EBV KHSV HIV
Obatoclax	direct inhibition	IAV
AT-101	direct inhibition	KHSV
A-1210477	direct inhibition	KHSV
Imatinib	downregulation of the MCL-1 level by the suppression of macrophage-colony stimulating factor (M-CSF) activation	HIV
Sorafenib + miR-193b	stimulation of the proteolytic degradation of MCL-1 (sorafenib) combined with the inhibition of *MCL1* mRNA translation (miR-193b)	HBV
Hydroxyanisole (BHA)	upregulation of the MCL-1 level by exerting the antioxidative effect in the cells treated with cisplatin	HBV
Resveratrol	downregulation of the MCL-1 level by the suppression of the signal transducer and activator of transcription (STAT)3 activation	HTLV-1
Interleukin (IL)-24	downregulation of *MCL1* mRNA translation through the induction of the endoplasmic reticulum (ER) stress	IAV

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
