# Peer review of "Mcl-1 Protein and Viral Infections: A Narrative Review"

_ijms, 2024, doi:10.3390/ijms25021138_

Round 1

Reviewer 1 Report

Comments and Suggestions for Authors

Title: Mcl-1 protein and viral infections

Reviewer:

Comments to the Author

1.     The "is" in line 33 is repeated.

2.     “The virus affecting the level of Mcl-1 includes” in line 41 has a grammatical error, “virus” should be used in the plural.

3.     The quotations should be added to this sentence “Research has suggested that the MCL-1 protein plays an important role in maintaining the viability of malignant cells during HCC development” in line 106.

4.     “suggests” in line 248 should be “suggest”.

5.     The quotations should be added to this sentence in line 251.

6.     The comma should be removed in the sentence “ It has been reported, that...” in line 366.

7.     Please try not to use phrasal verbs, such as “ sum up” in line 392, “ Taken together” in line 477.

8.     The expression of this sentence “Currently have been described seven species of human adeno- 438 viruses (HAdVs), named A-G, with 113 different genotypes that may contain multiple 439 genetic variants” in line 438-439 is incorrect, please correct it.

9.     “ This finding” in line 846 should be plural.

10.  “has” in line 979 should be “have”.

11.  Mcl-1 protein is closely related to viral infection, and it plays a role in immune response, antiviral infection, viral replication and antiviral therapy. The research progress can be briefly summarized in the introduction, and the importance of antiviral therapy can be highlighted at last.

12.  The structure and function of Mcl-1 protein are not introduced in this paper, so it is recommended to consult the literature and add this part.

13.  In the Conclusions, it is recommended to increase virus prevention and control strategies targeting Mcl-1 protein.

Comments on the Quality of English Language

No comments.

Author Response

We thank Reviewer #1 for his valuable time in providing feedback. We have accommodated all the Reviewer’s suggestions. The revisions made to the manuscript are described here and are also highlighted in the manuscript using yellow color.

Specific comments:

  1. The "is" in line 33 is repeated.

RESPONSE: The “is” in line 33 was deleted.

  1. “The virus affecting the level of Mcl-1 includes” in line 41 has a grammatical error, “virus” should be used in the plural.

RESPONSE: We used the plural form (“viruses”) in line 44.

  1. The quotations should be added to this sentence “Research has suggested that the MCL-1 protein plays an important role in maintaining the viability of malignant cells during HCC development” in line 106.

RESPONSE: We added the quotations to the aforementioned sentence in line 124.

  1. “suggests” in line 248 should be “suggest”.

RESPONSE: We replaced the “suggests” with “suggest” in line 265.

  1. The quotations should be added to this sentence in line 251.

RESPONSE: We added the quotation to this sentence in line 269.

  1. The comma should be removed in the sentence “ It has been reported, that...” in line 366.

RESPONSE: The comma was removed from this sentence in line 383.

  1. Please try not to use phrasal verbs, such as “ sum up” in line 392, “ Taken together” in line 477.

RESPONSE: “Sum up” was deleted from the text in line 409, and “Taken together” was removed from the fragments in lines 493 and 929.

  1. The expression of this sentence “Currently have been described seven species of human adeno- 438 viruses (HAdVs), named A-G, with 113 different genotypes that may contain multiple 439 genetic variants” in line 438-439 is incorrect, please correct it.

RESPONSE: The aforementioned sentence was divided into two shorter parts and reformulated into the following form: “Human adenoviruses (HAdVs) have been grouped into seven species, named A-G. So far, 113 different genotypes and multiple genetic variants have been described.”, in lines 454-456.

  1. “ This finding” in line 846 should be plural.

RESPONSE: We used the plural form (“These findings”) in line 861.

  1. “has” in line 979 should be “have”.

RESPONSE: We replaced the singular form (“has”) with the plural one (“have”) in line 993.

  1. Mcl-1 protein is closely related to viral infection, and it plays a role in immune response, antiviral infection, viral replication and antiviral therapy. The research progress can be briefly summarized in the introduction, and the importance of antiviral therapy can be highlighted at last.

RESPONSE: At the end of the introduction, we added a summary of the virological and immunological research devoted to the role of MCL-1 in viral infections. We also underlined the importance of searching for effective antiviral strategies. We replaced the ending sentence of the introduction: “Interestingly, MCL-1 down-regulation is observed and serves as a priming event for apoptosis during infections with various types of viruses, which is presented in detail in the following sections” with the extended fragment in lines 104-110.

  1. The structure and function of Mcl-1 protein are not introduced in this paper, so it is recommended to consult the literature and add this part.

RESPONSE: We added the recommended part in lines 54-64.

  1. In the Conclusions, it is recommended to increase virus prevention and control strategies targeting Mcl-1 protein.

RESPONSE: In conclusions, we extended the fragment summarizing the virus prevention and control strategies targeting Mcl-1 protein (in lines 1020-1038).

Reviewer 2 Report

Comments and Suggestions for Authors

To the authors:

Congratulations for your article.

This article is an excellent review of the MCL-1 protein and its role in cancer, as well as with certain viral infections that, could generate apoptosis or not. Without a doubt, this work alerts us as virologist about the interaction of MCL 1 and viral infections, how can influence the viral  infection cycle. The authors have carried out an extensive review on the topic, with an extensive bibliography that supports this review. In my opinion, the part of “MCL-1 protein in antiviral therapies and treatments of virus-associated diseases current knowledge and future perspectives” could be greatly reduced in its content. My only observation refers to figure 1 where the quality of the image could be improved (sharpness of the drawings and letters), in the legend it would be necessary to identify 1-that the figure is a mitochondria, 2-the indicator arrows (lines discontinuous or not) explain why they are different.
Excellent work.

Author Response

Congratulations for your article.

We thank Reviewer #2 for his valuable time in providing feedback and for congratulations. The revisions made to the manuscript are described here and are also highlighted in the manuscript by using the turquoise color.

This article is an excellent review of the MCL-1 protein and its role in cancer, as well as with certain viral infections that, could generate apoptosis or not. Without a doubt, this work alerts us as virologist about the interaction of MCL 1 and viral infections, how can influence the viral  infection cycle. The authors have carried out an extensive review on the topic, with an extensive bibliography that supports this review. In my opinion, the part of “MCL-1 protein in antiviral therapies and treatments of virus-associated diseases current knowledge and future perspectives” could be greatly reduced in its content.

RESPONSE: If possible, we would prefer to keep an expanded version of this part of the article. We believe that extensive information about the possible role of Mcl-1-targeting agents in antiviral therapies may be of interest to the readers.

My only observation refers to figure 1 where the quality of the image could be improved (sharpness of the drawings and letters), in the legend it would be necessary to identify 1-that the figure is a mitochondria, 2-the indicator arrows (lines discontinuous or not) explain why they are different.

RESPONSE: The quality of figure 1 was improved. The mitochondrion was labeled within the graphics. The (dis)continuity of indicator arrows was explained in lines 42-44:The continuity of the black arrows symbolizes the translocation of the given molecule(s) to the target structure(s), whereas the discontinuity means the lack (inhibition) of this event”.

Excellent work.

RESPONSE: Thank you very much. It is a great pleasure for us to read these kind words.

Reviewer 3 Report

Comments and Suggestions for Authors

Estimated Authors,

I've been invited to peer-review this narrative review on the topic of MCL-1 protein and viral infection. Through an accurate and very detailed approach, encompassing the role of MCL-1 in various viral infections, and its potential and/or documented interaction with several anti-microbial agents, Authors provide to their readers a very well written and documented point of view about this specific topic.

From my point of view, the present review has the only flaw of being a narrative review (please address this point in the main title, by changing it from "Mcl-1 protein and viral infections" to "Mcl-1 protein and viral infections: a narrative review"). Another potential limit, that authors could rapidly address in the later stages of the paper, is being their collected and conveyed evidence largely based on cell models rather than on "in-vivo" studies. Therefore, they cannot rule out (as the original studies course their reference to) that some of the results they have summarized would not be appreciable in clinical practice.

After these very limited shortcomings have been addressed, the paper could be accepted.

Author Response

Estimated Authors,

I've been invited to peer-review this narrative review on the topic of MCL-1 protein and viral infection. Through an accurate and very detailed approach, encompassing the role of MCL-1 in various viral infections, and its potential and/or documented interaction with several anti-microbial agents, Authors provide to their readers a very well written and documented point of view about this specific topic.

RESPONSE: We thank Reviewer #3 for his valuable time in providing feedback. The revisions made to the manuscript are described here and are also highlighted in the manuscript by using the green color.

From my point of view, the present review has the only flaw of being a narrative review (please address this point in the main title, by changing it from "Mcl-1 protein and viral infections" to "Mcl-1 protein and viral infections: a narrative review").

RESPONSE: We changed the title from "Mcl-1 protein and viral infections" to "Mcl-1 protein and viral infections: a narrative review".

Another potential limit, that authors could rapidly address in the later stages of the paper, is being their collected and conveyed evidence largely based on cell models rather than on "in-vivo" studies. Therefore, they cannot rule out (as the original studies course their reference to) that some of the results they have summarized would not be appreciable in clinical practice.

After these very limited shortcomings have been addressed, the paper could be accepted.

RESPONSE: We agree with Reviewer #3. At the end of the conclusion (lines 1043-1045), we addressed the limit of the current knowledge and underlined the need for further (in vivo) studies. We added the following fragment:

“So far, the antiviral activity of MCL-1-targeting agents has been examined mainly in cell models. Thus, further extensive investigations are needed to confirm the promising initial findings”.

Round 2

Reviewer 1 Report

Comments and Suggestions for Authors

No